# Exploring the perspectives of healthcare workers and Program managers on the use of Truenat as a new tool for TB and DR-TB diagnosis in Nigeria: A qualitative study

Nkiru Nwokoye[1], Austin Ihesie[2], Jamiu Olabamiji[3], Kingsley Ochei[4], Rupert Eneogu[2], Michael Umoren[1], Femi Odola[3], Debby Nongo[2], Aderonke Agbaje[3], Bethrand Odume[1], Omosalewa Oyelaran[2], Wayne van Germert[5], Lucy Mupfumi[5], Elom Emeka[6], Chukwuma Anyaike[6], Sarah Cook Scalise[7], Edmund Ndudi Ossai[8]*

**1** KNCV Nigeria, Abuja, Nigeria, **2** HIV AIDS & TB Office, USAID Nigeria, Abuja, Nigeria, **3** IHV-Nigeria, Lagos, Nigeria, **4** USAID Leap Project, Abuja, Nigeria, **5** STOP TB Partnership, Geneva, Switzerland, **6** Federal Ministry of Health, Abuja, Nigeria, **7** Infectious Disease Office/Tuberculosis Division, Global Health Bureau, USAID, Washington, Washington, D.C., United States of America, **8** Department of Community Medicine, College of Health Sciences, Ebonyi State University, Abakaliki, Nigeria

* ossai_2@yahoo.co.uk

**Data Availability Statement:** All relevant data are within the paper and its Supporting Information files.

## Abstract

### Background

World Health Organization in the year 2020 recommended the use of Truenat as a replacement for smear microscopy in Tuberculosis (TB) diagnosis and detection of rifampicin resistance. This study was designed to assess enablers and barriers to effective implementation of Truenat assays for TB diagnosis in Nigeria and determine the acceptability of use of Truenat among healthcare workers and TB Program managers in Nigeria.

### Methods

A descriptive exploratory study design was used. Qualitative data were collected via Zoom platform using a pre-tested focus group discussion (FGD) guide and key informant interview (KII) guide. Four FGDs were conducted among Truenat laboratory staff, State Quality Assurance Officers, Local Government Tuberculosis Supervisors and Clinicians working at Truenat sites. Three KIIs were conducted among laboratory leads of Truenat implementing partners and the National TB Control Program.

### Results

All the participants attested to the reliability and acceptability of Truenat results, they also highlighted the portability and ease-of-use especially for community outreach testing. Stakeholder engagement, training of Truenat laboratory staff and the perceived low operational cost associated with Truenat were the enablers of Truenat implementation. Major barriers to the implementation included human resource for health constraints, lack of power supply, frequent Trueprep breakdown and Truenat machine connectivity. The same human

**Funding:** The study was funded by the United States Agency for International Development (USAID) through the Stop TB partnership introducing new tools project (iNTP). The USAID Country Team in Nigeria was involved in the design and implementation of the study and in producing the manuscript. However, the views as expressed in the manuscript do not represent that of USAID.

**Competing interests:** The authors have declared that no competing interests exist.

**Abbreviations:** TB, Tuberculosis; DR-TB, Drug resistant tuberculosis; FGD, Focus Group Discussion; HIV, Human Immunodeficiency Virus; IHVN, : Institute of Human Virology Nigeria; iNTP, Introducing New Tools Project; KII, Key Informant Interview; LON, Local Organization Network; NHREC, National Health Ethics Research Committee; NTBLCP, National Tuberculosis, Leprosy and Buruli ulcer Control Program; RIF, : Rifampicin; SQAO, State Quality Assurance Officer; USAID, United States Agency for International Development; WHO, World Health Organization.

resource constraint was viewed as the major barrier to scale up of Truenat while employment and retention of Truenat staff coupled with training were seen as the facilitators to scale-up. The participants implored the manufacturers of Truenat to increase the number of modules for Truenat, enable the use of stool for diagnosis of TB and attach a solar panel to the machine.

## Conclusion

Truenat has gained high acceptance among health workers and TB Program managers in Nigeria. The perceived low operational cost and low infrastructural requirements have been a major boost. There is a need to ensure the retention of health workers especially the Truenat laboratory staff. Training should be sustained including the payment of incentives. Increasing the number of modules of the Truenat machine, enabling the use of stool for TB diagnosis and having a solar panel attached to Truenat machine are essential.

## Introduction

In 2021, tuberculosis (TB) was the second leading infectious cause of death after COVID-19 and is the leading cause of death among people with HIV. It remains a global epidemic with an estimated 10.6 million people falling ill with TB in 2021 with 450,000 of the cases being due to rifampicin resistant TB [1]. Majority of the estimated 4.2 million "missing" cases who were either undiagnosed or unreported to public health authorities are from developing countries. Nigeria is one of the ten countries that contributes to the highest burden of missing TB cases globally and this is due to limited access to TB diagnosis. The country accounts for 4.6% of the global TB burden and also has a high triple burden of TB, DR-TB and HIV associated TB [2].

The World Health Organization (WHO) is advocating increased access to early and accurate diagnosis using a molecular WHO-recommended rapid diagnostic test. However, reduction in TB-related morbidity and mortality in resource-limited countries continues to be impeded by poor access to rapid and cost-effective diagnostic tests due to high infrastructural and operational costs of the widely available PCR tests [1]. For several years, AFB microscopy was the mainstay of TB diagnosis in Nigeria. The use of the microscopy test was faced with several challenges including the inability to diagnose Rifampicin resistant TB. In the year 2010, the WHO recommended the molecular GeneXpert MTB/Rif assay as the initial diagnostic test for TB. The implementation of the GeneXpert MTB/Rif assay was also limited because of the challenges associated with infrastructural and environmental requirements of the GeneXpert machine, Nigeria being a developing country.

There has been significant global research and investment in the development of cost-friendly, highly sensitive PCR platforms that help address the challenges of access for resource-constrained settings [3]. Thus, the WHO in the year 2020 recommended the use of one of the new diagnostic tools, Truenat, as a replacement for smear microscopy in Tuberculosis (TB) diagnosis and detection of rifampicin resistance. This followed its rapid communication on the use of molecular assays as initial tests for the diagnosis of Tuberculosis [4].

Truenat is a portable, battery-operated, chip-based, rapid molecular test manufactured by Molbio, India, which detects Mycobacterium tuberculosis (MTB) in approximately one hour and rifampicin (RIF) resistance in another 40–60 minutes. The Truenat instruments function optimally at room temperature and up to 40 degrees Celsius. It is a cost-effective molecular

test with a sensitivity and specificity comparable to GeneXpert, that could be used as point-of-care or near point-of-care test in peripheral facilities and "low-infrastructure" settings to diagnose TB and potentially save thousands of lives [5]. Truenat can also be operated with no requirement of skilled manpower [6]. The deployment of Truenat in Andhra Pradesh, India as a point of care diagnostic test revealed that Truenat improved identification of individuals with newly diagnosed TB and was operationally feasible under the revised national tuberculosis control program [5]. Also, Truenat, while used at a point of care in India, improved linkage to care [7]. This could be achieved by reducing sputum referrals to GeneXpert sites which has the added advantage of decreasing turn-around time and time to initiation of TB treatment. Due to its low operational cost, the cost efficiency of Truenat when compared to GeneXpert or smear microscopy was included as an additional advantage [7].

Consequent to the recommendation of Truenat by the World Health Organization for TB diagnosis and detection of rifampicin resistance, the Stop TB Partnership in 2021 through the USAID-funded *introducing New Tools Project* (iNTP), donated 38 Truenat systems to Nigeria to expand access to high quality, innovative diagnostic tools for TB among populations in hard-to-reach areas. This study sought to explore the operational feasibility of the use of the Truenat machines for TB and drug-resistant TB (DR-TB) diagnosis in Nigeria. This study was also designed to assess the enablers and barriers to effective implementation of Truenat assays for TB diagnosis in Nigeria and determine the acceptability of the use of Truenat among healthcare workers and TB Program managers in Nigeria.

## Methodology

### Study setting

The study was conducted in 14 states in Nigeria implementing the TB Local Organization Network (LON) project, where the 38 Truenat machines from the USAID-funded *introducing New Tools Project* has been installed. The TB LON is a USAID-funded project where USAID supports local organizations in TB-priority countries to implement locally generated solutions to improve TB diagnosis, treatment, and prevention services. The LON project in Nigeria is categorized into 3 regions–LON 1, 2, and 3. Kano, Kaduna, Katsina, Bauchi, Taraba, Plateau and Nasarawa make up the LON 1 states while Anambra, Imo, Delta, Akwa Ibom, Rivers, Cross River and Benue are the states in LON 2 and Lagos, Oyo, Ondo and Osun are in LON 3 States. KNCV Nigeria was awarded the LON 1 and 2 regions while Institute of Human Virology, Nigeria received the LON 3 award. The states selected for the study included Kano, Katsina, Kaduna, Bauchi, Nasarawa and Taraba in the northern part of the country as well as Akwa-Ibom, Anambra, Cross River, Delta, Rivers, Lagos, Osun and Oyo States in the southern part of the country. Historically, these states were known to experience varying levels of diagnostic gaps for TB evaluation hence they were selected for implementation of Truenat under the iNTP. Three of the states (Lagos, Osun and Oyo) in the southwest geo-political zone of the country are part of the TB-LON 3 project implemented by IHVN, while the remaining eleven states are under TB-LON 1&2 projects implemented by KNCV Tuberculosis Foundation Nigeria.

### Study design, participants and sampling

This was a qualitative descriptive study. The study population included healthcare workers involved in the implementation of Truenat from the study sites and TB Program managers who are involved in the Truenat implementation. Thirty-three participants took part in the study. They included 30 healthcare workers who participated in four focus group discussions (FGDs) and three program managers who participated in key informant interviews (KIIs).

Four groups of healthcare workers were included in the FGDs namely, nine laboratory personnel operating the Truenat machines, nine Tuberculosis Supervisors in Local Government Areas implementing Truenat, seven Facility level Clinicians and eight State Quality Assurance Officers (QAOs). The three program managers who participated in the KIIs included the laboratory leads of the implementing partners and the laboratory lead of the National Tuberculosis, Leprosy and Buruli ulcer Control Program. No individual contacted to take part in the study declined the offer. Data saturation was reached after the interviews where no new information as obtained. This was because the different groups of health workers and the mangers approached the study from their unique but different roles and contributions to the TB program in Nigeria.

## Inclusion and exclusion criteria

Purposive selection was used in the selection of participants for the study across the 38 Truenat sites in the country. Criteria for selecting the study participants included geographical spread, coordinating partners in Truenat implementing facilities, previous or current knowledge of GeneXpert operation, previous participation in a virtual meeting, gender, and willingness to participate in the study. No interview was repeated or inability to participate in the interview twice.

## Study instrument and data collection method

Information was obtained from the participants using pre-tested focus group discussion and key informant guides. The pre-testing of study tools was done virtually among healthcare workers who had previously worked on Truenat sites. The aim of the pre-testing was to identify and correct ambiguities of the study instrument. The FGDs and KIIs took place virtually via Zoom with the help of Zoom links which were specifically created for the purpose of the discussions. All discussions were recorded electronically via Zoom and the interviewer made notes during the interview. Prior to group meetings, participants were contacted (via messaging, WhatsApp and telephone calls) to agree on convenient dates and time for the group discussions. The interviews were conducted in English language. The average duration of the FGDs was one hour twenty minutes while for the KIIs, the average duration was fifty-five minutes. The participants in the KIIs had the transcripts of the interviews returned to them for comments and they also provided feedbacks.

## Data management

The recorded discussions of FGDs and KIIs were transcribed verbatim immediately after each session. The script was checked against the recording by an independent reviewer. As a way of verifying the quality of transcriptions, the recordings were doubly transcribed after which both scripts were checked for similarities and where differences existed they were reconciled by the independent reviewer. Coding of transcripts was done by two researchers based mostly on predetermined themes, in some cases themes were created as they emerged during the coding process. This was followed by categorization of data into themes. The themes from each interview were reviewed by the researchers and grouped under wider themes and sub-themes (Table 1). NVivo statistical software version 14 was used in the analysis of data [8].

## Ethical considerations

Ethical approval to conduct the study was granted by the National Health Ethics Research Committee, (NHREC). (Approval number NHREC/01/01/2007-05/04/2023). The recruitment

**Table 1. Themes, sub-themes and codes generated from the study.**

| s/n | Themes | Sub-themes | Codes |
|---|---|---|---|
| 1. | Perception of use of Truenat | | Generally, Truenat is user friendly, if you can operate an android gadget, you can operate Truenat |
| 2. | Enablers and barriers to use of Truenat and scale-up | Enablers | Stakeholders' engagement is what served as enabler and the engagement was all encompassing. |
| | | Barriers | Truenat sites are in hard-to-reach areas and we found out that we had human resource constraints in some of the sites |
| 3. | Operational advantages and disadvantages of Truenat | Advantages | Comparing with TB-LAMP, you know TB-LAMP does not detect RIF resistance, it only tells you whether there is TB or not. |
| | | Disadvantages | Another disadvantage of Truenat is that if you detect TB, you have to run a test for RIF resistance, you will have to work on the sample again unlike GeneXpert |
| 4. | Use of Truenat in community-based active case finding | Involvement of Truenat in active case finding | Truenat could be used in a mobile fashion unlike GeneXpert. We have actually used that on what we call 'Wellness on Keke |
| | | Challenges in using Truenat for active case finding | A lot of time is spent packing and unpacking the Truenat machine for community outreaches |
| 5 | Reliability and acceptability of Truenat test results. | Reliability of Truenat test results | The reliability of the Truenat test is just okay. I will say it is the same thing when compared to GeneXpert. |
| | | Acceptability of Truenat test results | In my state of operation, the results obtained from Truenat are well accepted since they were informed before-hand that it works like GeneXpert. |
| 6. | Facilitators and barriers to scale up of Truenat | Barriers to scale up | There must be someone, a laboratory staff who is capable of conducting the test. If there is no one to operate the Truenat machine, it is a barrier |
| | | Facilitators of scale up | Does the laboratory have the things needed for Truenat machine to take off in the facility. |
| 7. | Improving Truenat: advice to Manufacturers | | I will also advise them that they should validate Truenat using other specimen because the way to go now is the use of specimen that is easy to get like stool" |

of participants for the study started from April 11th to April 28th 2023 after which data collection processes started. The study lasted for one month. A written and signed informed consent was obtained from each participant in the study while assuring them of confidentiality and anonymity of data. The respondents were also informed that participation in the study was voluntary and they were assured that there would be no victimization of respondents who refused to participate or who decided to withdraw from the study after giving consent.

## Interviewer characteristics

The interviewer is a Senior Lecturer in the Department of Community Medicine of a Nigerian University. He has been involved in some qualitative studies [9, 10]. The interviewer was introduced to all the participants in the study and his role in the research was explained to them.

# Results

## Participants' profile

The age of the participants ranged from 30 years to 54 years with a median age of 44 years. More than half of the participants, 54.5%, were males. Majority of the participants, 81.8% have attained tertiary education. More than two thirds of the participants, 68.5%have worked in the National TB Program for more than five years.

## Themes

Six themes emerged from the analysis, including, perception of use of Truenat, enablers and barriers to implementation and scale up, operational advantages and disadvantages of Truenat,

**Fig 1. Emerging themes on operational feasibility and implementation of Truenat in Nigeria.**

use of Truenat in community-based active case finding, reliability and acceptability of results from Truenat, and improving Truenat: advice to manufacturers (Fig 1).

## a). Perception of use of Truenat

All the participants expressed positive views about Truenat TB tests and acknowledged their effectiveness in TB diagnosis. The attributes emphasized by the participants included the test being battery-operated, highly sensitive (including its ability to detect RIF resistance), and its usefulness in hard-to-reach areas as reasons for their positive perception of the tool. They highlighted the platform's user-friendly interface, its portability and ease of use, which makes it suitable for point-of-care testing. All the participants placed the Truenat machine at par with the GeneXpert but for a few of the participants Truenat has a comparative advantage. One of the participants summarized his view this way:

*"Generally, Truenat is user friendly, if you can operate an android gadget you can operate Truenat. The result from the Truenat test is accurate, precise and reliable. Also, it detects RIF (referring to rifampicin) Resistance.* (Participant, L4).

The participants mostly placed emphasis on areas where Truenat appears to have an edge over the GeneXpert. One of the participants pointed out its mode of operation which eventually made it very relevant to the study area, an increase in the identification of newly diagnosed TB patients.

*"Truenat doesn't require a whole lot of infrastructural upgrades before you can start using it. So, that makes it easy for us to get to those hard-to-reach areas and test."* (Participant K1).

In effect, the location of the Truenat machines at peripheral facilities with limited access to mWRDs reduced the pressure on the sputum referral system and GeneXpert laboratories that receive high and overwhelming volume of samples due to high population coverage.

*"Like in our state, we have three Local Government Areas where Truenat machines are situated, so samples are no longer transported away from these Local Government Areas. They now conduct their TB tests and they get their results on time, within twenty-four hours."* (Participant Q3).

Likewise a laboratory staff notes there was a decrease in turn-around time.

*"The turnaround time is very good, the highest turn-around time my facility has given is 48 hours. I do whatever I have to do and not allow the samples to exceed 48 hours in my hand without processing it."* (Participant L2).

Attesting to the suitability of Truenat for our country context a provider noted:

*"Truenat is made for us where we don't have electricity and where the temperature does not affect the machine as it does to GeneXpert. It has been an alternative source of testing for us especially as it has shown to give the same value like GeneXpert. "(Participant* C6)

In conclusion, Truenat is a relief and a support to GeneXpert in Nigeria.

*"It has been wonderful having Truenat in my Local Government Area. Formerly, we had only one GeneXpert machine. What happened is that by taking all the samples to GeneXpert, sometimes there will be a lot of backlog."* (Participant S5).

### b). Implementation of Truenat (Enablers and barriers to use and scale-up)

*i). Enablers.* Certain factors served as enablers to the successful implementation of Truenat in Nigeria. All the participants agreed that stakeholder engagement was key to adoption and subsequent successful implementation of Truenat at national and subnational levels. Participants emphasized these points in the following statements.

*"Stakeholders' engagement is what served as enabler and the engagement was all encompassing. It was from top to bottom, from NTBLCP to the community people to USAID and the implementing partners and all were ready to embrace the new tool."* (Participant K2).

*"In my facility, there was a sensitization programme that was carried out during the installation of our Truenat machine, the health workers in neighbouring health facilities were all invited. We introduced the Truenat machine to them and told them how accurate the result from the machine could be."* (Participant L5).

Two participants who are Local Government Supervisors asserted that if there is no proper sensitization of health workers on Truenat that could hinder the scale-up of the diagnostic tool in Nigeria. Following stakeholder engagement, the next enabling factor in the implementation was training of health workers especially the laboratory staff.

*"We ensured that those who were involved in the use of Truenat were trained. This massive training approach for all laboratory staff helped in the successful implementation of Truenat."* (Participant K3).

In addition to monitoring utilization, the New Tools implementing partners closely monitored errors and machine challenges across the sites. This provided opportunities for further training during site visits as was corroborated by a laboratory staff:

*"There has been prompt training of people using the Truenat machine. Since we got Truenat, we have had training two or three times which helped us to get more versatile with the working of the machine."* (Participant L1).

The next enabling factor in the Truenat implementation was the low operational cost associated with Truenat. All the participants agreed that the operational cost was moderate because of the low infrastructural demands of Truenat. This was because Truenat was less dependent on environmental conditions such as temperature and hence did not require constant electricity and air-conditioners in comparison to Gene Xpert.

*"The atmospheric condition under which Truenat works is very friendly. It doesn't need much electricity to work or structures like air conditioners. It requires minimal infrastructure in a room where there is a bench and minimal electricity,"* (Participant Q7).

A participant however remarked that the Truenat machines and the reagents were provided as a gift to the country by an international agency and the implementing partners focused on upgrading the laboratories and other activities.

*ii). Barriers.* The participants also identified several barriers encountered during the implementation of Truenat for TB diagnosis in Nigeria. Key barriers identified included human resource challenges, lack of power supply and Truenat machine related problems. Of these, availability of human resources was the most significant barrier during the Truenat implementation program. Specific to note was the lack of personnel to handle the machine. The participants exemplified their thoughts thus:

*"Again, we found out that those Truenat machines because their locations or the desired locations were to be at the peripheral laboratories where we had neither GeneXpert nor TB LAMP, those sites that are in hard to reach areas, we found out that we had human resource constraints in some of the sites. So, for some of those sites, we supported them with human resource"* (Participant K2).

*"The only barrier I see in the use of Truenat for TB diagnosis in my area is irregular power supply."* (Participant L7).

*"We have had discussions about the Truenat machine with the donor agency on the possibility of installing solar panel that will be supplying power to the machine as the main source so that power supply will be constant. The public power supply can then serve as an alternative source."* (Participant C3).

The challenge of poor power supply necessitated the provision of funds by the implementing partners for the fueling of generators but ultimately the plan is to provide solar panels for the Truenat machines. For laboratory personnel, lack of consumables was seen as the greatest hindrance to the use and scale up of Truenat. One of the participants pointed out that even though there has never been a shortage of consumables in his facility, he recalled there was a time the kits expired and that caused a downtime.

*"The only barrier that we can actually have is if there is no available kit (reagents) to work with. However, there has not been any moment we don't have kit to work with except recently when the kit expired and we had to put work on hold."* (Participant L2).

The next major barrier to the successful implementation of Truenat for TB diagnosis in Nigeria was the Truenat machine's connectivity. Participants suggested that the challenges impacting connectivity have yet to be resolved.

*"The barrier that comes to mind is connectivity-linking up Truenat to digital connectivity network. We are still not yet there. Though a lot has gone into that; but I believe that when we get there, implementation will improve much more."* (Participant K2).

In addition, a few of the participants mentioned that the frequent breakdown of TruePrep equipment was also a major challenge to implementation.

### c). Operational advantages and disadvantages of Truenat

*i). Advantages.* Most of the participants highlighted the minimal infrastructural requirement of the Truenat machine as a key advantage of the Truenat over GeneXpert. This may be due to the lessons learnt with the introduction of GeneXpert in the country. Truenat machines do not need constant electricity or an air conditioner to function. As a result, Truenat has become a very relevant diagnostic tool in rural areas of the country where power is in short supply.

*"GeneXpert is not cumbersome but these Truenat machines are wonderful, in fact one of them is installed in one of our hard to reach areas and it's the machine we are using to detect TB in the Local Government. It has been helpful.* (Participant Q2).

All the participants were specific in pointing out the reasons why Truenat is preferable over TB-LAMP. Major reason highlighted was the ability of Truenat to detect Rifampicin resistance

*"Comparing with TB-LAMP, you know TB-LAMP does not detect RIF resistance, it only tells you whether there is TB or not. If there is, you have to send it to the nearest site to ascertain the RIF status. (referring to RIF Resistance)."* (Participant L3).

*ii). Disadvantages.* The key operational disadvantages of the Truenat are limitations in the type of samples used, throughput, the many steps involved in diagnosis, the lack of solar panels, and being less durable than TB-LAMP. There was also an observation that when the sputum volume is small, the GeneXpert could detect mycobacterium TB while Truenat may

not. One of the participants explained the difference between using Truenat and GeneXpert in detecting RIF resistance, noting that GeneXpert has an advantage over Truenat.

*"Another disadvantage of Truenat is that if you detect TB, you have to run a test for RIF resistance, you will have to work on the sample again unlike GeneXpert."* (Participant Q6).

There are also differences in the manipulation of the machines in that Truenat was remarked to require a lot of hands on, a lot of steps when compared with GeneXpert. Perhaps because of the relevance of childhood TB diagnosis in Nigeria, the inability of the Truenat machine to use stool samples for TB diagnosis was assessed by the participants to be a significant challenge. The importance of the ability to support childhood TB diagnosis was further highlighted in a participant's comment drawing the attention of the manufacturers to this major disadvantage:

*"Truenat can only process sputum sample. In this month of May, there is going to be child-hood TB week and Truenat will not be used because majority of the samples that will be collected will be stool for testing TB. So it is a major disadvantage and should be looked into."* (Participant Q4).

The number of modules available for testing using Truenat is limited when compared with GeneXpert and TB LAMP. This is because the Truenat machine that is presently in Nigeria has only two modules thus limiting the number of tests that could be conducted at one time.

*"The disadvantage of Truenat is that in a typical day, you can only conduct 8–12 tests. This number is not encouraging when compared to the workload in a day which could be between 50 to 100 samples."* (Participant L2).

There was also an assertion that TB-LAMP is more durable and stronger than Truenat hence an advantage over Truenat. In addition, it was reported that Truenat does not come with a solar panel as part of its package, and that was considered a disadvantage also.

*"Truenat does not come with 'a' solar panel package while TB-LAMP has that package and it is actually helping us in hard-to-reach areas where we don't have electricity. We use TB-LAMP in outdoor testing without charging unlike Truenat."* (Participant Q6).

Countering some of the perceived disadvantages, a participant redefined some of the noted operational disadvantages of Truenat as advantages that could be countered. On the issue of detecting mycobacterium tuberculosis first before testing for Rifampicin resistance, it was pointed out that one will be able to test more samples once the test is negative. Regarding True-nat requiring a lot of hands-on and manual manipulation for sample processing unlike Gen-eXpert, the participant explained that after a period of time, the laboratory staff eventually get used to the many steps involved in the Truenat test and adapt to them.

Similarly, another participant believed there were no disadvantages with the use of the Truenat machine for TB diagnosis. In her words: *"I have not been able to think of anything too major on the aspect of Truenat to be called a disadvantage"* (Participant L3). She emphasized that even though there is no evidence yet on the use of Truenat to test for TB in stool samples that it is a work in progress and that soon that will be taken care of through a planned USAID study. She also looks forward to a Truenat with many modules. Likewise, there is an expectation that the updated model will provide solar panels.

### d). Use of Truenat in community-based active case finding

*i). Involvement of Truenat in active case finding.* Regarding the use of Truenat for active case finding in the communities, all participants identified the Truenat platform as an efficient and beneficial way to scale up community-based active case finding for TB. *"Truenat could be used in a mobile fashion unlike GeneXpert. We have actually used that on what we call 'Wellness on Keke'. This is a new innovation by KNCV* (an NGO) *where we deploy the tricycle (ie the keke); with PDX (ie the Portable Digital X-ray) and we couple that with Truenat and we take them to the field for TB diagnosis."* (Participant K2).

Participants emphasized the convenience and speed of obtaining test results at the point of care, allowing for immediate treatment initiation. Truenat was further commended for its ability to work in environments without electricity or air-conditioner. *"I have worked outside the facility with Truenat machine for community case finding of TB. We streamline our presumptives with the help of the X-ray. We test with Truenat and receive our results at the field and because the DOTS providers are on ground, the patients are placed on treatment immediately. We could also do contact tracing same time."* (Participant L4).

There were some community outreaches that were conducted with only the Truenat machine.

> *"We usually take the Truenat machine to the field during outreaches and in the field as we know, it requires that minimal samples are tested and for cases that are positive, treatment is commenced immediately.* (Participant Q4).

*ii). Challenges in using Truenat for active case finding.* However, there were challenges in using Truenat platform for community interventions. The first limitation identified was related to limitation in the volume of simultaneous sample that the Truenat can run. Another challenge is the time required for packing and unpacking of the machine for outreaches. The agencies minimized the issue around packing/unpacking by transporting the Truenat machine to the outreach area a day before. To overcome both challenges, a participant envisaged a situation when Truenat with a particular portable digital X-ray will be kept permanently for community interventions while also hoping for an increase in the number of modules for the Truenat machines.

### e). Reliability and acceptability of Truenat test results

*i). Reliability of Truenat test results.* All the participants had no doubts about the reliability of results from the Truenat machine. This was because of the notion that the results obtained from Truenat are comparable to that of GeneXpert. Also, most of them were aware that the World Health Organization has approved the use of Truenat for the diagnosis of TB. Moreover, Truenat has been included in the national TB guidelines and the health workers were aware that the laboratory staffs have been trained on the use of Truenat machine. One of the Clinicians summed up the thoughts of the participants this way:

> *"I am confident using Truenat results since it is MTB/RIF test, is gene-based and its sensitivity is as accurate as GeneXpert. I am very confident using the test results."* (Participant C6).

One of the participants, a Quality assurance officer explained the process of testing for TB using Truenat and comparing it with the Smear microscopy method as a way to justify the reliability of the result He had his to say:

*The reliability of the Truenat test is just okay. I will say it is the same thing when compared to GeneXpert. It is very reliable because it is same way that GeneXpert detects MTB and then RIF Resistance that Truenat operates. It is very reliable.* (Participant Q6).

A laboratory personnel who used to carry out smear microscopy went further to describe the processes involved in Truenat testing as a pointer to the reliability of its results

*"Truenat has to deal with DNA extraction which makes it more reliable. This is not the same as when we work on Smear microscopy method which we have been using before."* (Participant L7).

Another Quality assurance officer provided information that they usually conduct quality control checks on the Truenat machine which necessitates that its results should be reliable. A laboratory staff also asserted that once the standard operating procedures of Truenat or any other test is followed that the result of that test will be reliable. All these facts not-withstanding, some of the participants convinced themselves that Truenat test results are reliable by validating the results using GeneXpert. One of the participants expressed his view this way:

*"Truenat results are very reliable. What I did apart from the quality control of the machine was that before running the samples; some quantities were taken to Truenat sites, same samples were also taken to GeneXpert sites just to be very sure that we are getting good results. And the two always came out with the same result showing that these Truenat results are reliable."* (Participant Q3).

*ii). Acceptability of Truenat test results.* All the participants also acknowledged the fact that the other health workers were confident of the results from Truenat machine, although some were doubtful at the beginning of the process. The acceptance of the result of Truenat by other health workers was attributed to efforts at sensitization.

*"Also, during the initiation of Truenat in my LGA, most of the other health workers were invited and the information concerning Truenat was passed to all. So, most of the other health workers knew about Truenat already. They know the results are reliable."* (Participant S4).

These attempts at validating the results from the Truenat machine eventually helped in increasing the utilization of Truenat sites. This was because the health workers became convinced on the reliability of Truenat results. To a large extent, GeneXpert was used as the reference while introducing Truenat to the people.

*"In my state of operation, the results obtained from Truenat are well accepted since they were informed before-hand that it works like GeneXpert. Based on this, they do not doubt the results from Truenat, they considered the results very reliable."* (Participant L5).

A participant has an impression that the supply of a quattro Truenat machine may make the people see 'GeneXpert' in Truenat.

*If 'quattro' Truenat machines are supplied the healthcare workers will see the Truenat machine as being the same as GeneXpert machines."* (Participant Q1).

One of the participants who is a laboratory personnel however made a differentiation among the health workers and their acceptance of the Truenat test results. She was of the opinion that the Riders for Health preferred sending samples to GeneXpert sites even when there is a nearby Truenat site. It could be that the relevance of their assignment was dependent on the number of positive samples delivered hence they preferred sites where the chances of a positive result was very high. Suffice it to say that whenever there is doubt with regards to the quality of Truenat test result, the GeneXpert is always available for the purpose of validation via specimen referral.

### f). Facilitators and barriers to scale up of Truenat

*i). Barriers to scale up.* Most of the participants believed human resource constraints will be the biggest barrier to the scale up of Truenat to other LGAs and health facilities in Nigeria. The focus on human resource is the lack of laboratory staff to operate the Truenat machine.

*"There must be someone, a laboratory staff who is capable of conducting the test. If there is no one to operate the Truenat machine, it is a barrier."* (Participant S3).

There is also a request by the laboratory staff for the payment of incentives. One of the participants had this to say:

*"From my side, the laboratory staff are always shouting for incentives, they have been crying that they are the least supported healthcare workers as in financial support."* (Participant C2).

One participant however tied human resource and space constraints in health facilities as acting together hence a very big factor. For the participants who were laboratory personnel, lack of consumables were seen as the greatest hindrance to the scale up of Truenat. One of the participants pointed out that even though there has never been a shortage of consumables in his facility, he recalled there was a time the kits expired and that caused a downtime.

A few of the participants pointed to irregular power supply as the greatest barrier to the effective use of Truenat.

*"The only barrier I see in the use of Truenat for TB diagnosis in my area is irregular power supply."* (Participant L1).

One participant being particular on his state of operation pointed to security concerns as a very big limitation to the scale up of Truenat to other LGAs or health facilities. The participant expressed his opinion this way:

*"In my state, we have security problems. Some local government areas in my state (about five or six) with high TB burden and high sample load have security problems and because of that we cannot install Truenat machines in those areas."* (Participant Q2).

One laboratory staff however viewed poor quality sputum as the biggest barrier to effective use of Truenat tests. He was of the opinion that those who supervise sputum production by clients should ensure that the sputum produced is of good quality and sufficient.

*ii). Facilitators of scale up*. Participants' views on facilitators on the scale up of Truenat had little semblance with the pattern of identified barriers. However, issues related to human resource for health received priority. These issues included the need to employ more health workers, (especially the laboratory staff), training and motivation of the health workers especially those in charge of Truenat machines.

*"Health workers, especially the laboratory staff, those that will be handling the Truenat machine should be employed."* (Participant S7).

The participants also placed emphasis on training all laboratory staff in any facility where Truenat will be located. As a stop gap measure, they advised the implementing partners to recruit ad-hoc staff in facilities where laboratory staff are not available or are few in number. There was also a strong call on the part of the participants that incentives should be provided to laboratory staff who handle the Truenat machines.

Surprisingly, the laboratory staff did not pay any attention to human resource for health in their projections on barriers and facilitators to scale up Truenat in the country. Most of the laboratory staff were however concerned with increasing the awareness of TB among the people as a facilitator to the scale up of Truenat. A few of the participants who were Quality Assurance Officers were concerned that lack of basic laboratory bench infrastructure at most primary health care laboratories could be a barrier to effective scale up as infrastructural upgrades would be required.

*"You know if you want to scale up Truenat machine to other Local Government Areas, you have to consider the facility first or the laboratory. Do they have the things needed for this machine to take off in the facility.*" (Participant Q7).

While the laboratory personnel were focusing on increasing awareness of TB among the populace, a few of the Local Government Supervisors focused on adequate advocacy and engagement of stakeholders in each LGA before Truenat could be introduced there.

Two of the participants who were TB Supervisors hinted that the advantages recorded so far for the Truenat machine since its introduction in the country should facilitate the scale-up of its use. From each group that participated in the discussions came voices that passionately appealed that Truenat should be made to detect TB using stool samples. One of the TB Supervisors had this to say,

*"If it can be possible for the Truenat machine to be checking presumptives using stool sample, it will help us more. "*(Participant S5).

One participant who was a Quality assurance officer was very sure that the use of Truenat will indeed be scaled up in the country. She was convinced that the importance of using molecular tests which detects RIF Resistance at diagnosis will ensure that the scale up of Truenat will be inevitable.

## g). Improving Truenat: Advice to manufacturers

The participants made suggestions and recommendations on possible ways to address the identified barriers hindering the smooth implementation of Truenat in the country. These included increase in the number of modules for the Truenat machine, use of stool for the

diagnosis of TB, production of detachable batteries and solar panels, making spare parts available and awareness creation.

All the participants wanted the number of modules in the Truenat machine to be increased

*". . . ..to look into production of a higher module machine that can actually test more. So if you have like a machine that has up to 20 or 30 modules, you will be able to cover a whole lot of ground."* (Participant K3).

With a focus on diagnosis of childhood TB, all the participants wanted the use of stool samples to test for TB using the Truenat machine. In-fact all the laboratory staff who participated in the discussion expressed their willingness to use stool to test for TB using Truenat once that approach is approved by relevant authorities.

*"I will also advise them that they should validate Truenat using other specimen because the way to go now is the use of other specimen (especially for children) that is user-friendly and is easy to get like stool"* (Participant K1).

The participants also pointed out the importance of having a solar panel attached to the Truenat machine just like the TB LAMP. Another approach towards improving the life span of the Truenat machine was for the battery attached to the Truenat machine to be detachable instead of being in-built. There was a focus on the functionality of the Truenat machine like improving the technology of Trueprep. Another important recommendation by some of the participants was the need to make Truenat spare parts available in the country.

## Discussion

This study sought to explore the operational feasibility, enablers and barriers to the effective implementation of Truenat assays for TB diagnosis in Nigeria and also determine the acceptability of the use of Truenat among healthcare workers and TB Program managers in Nigeria. All the participants in the study affirmed that the results of Truenat tests are reliable, hence acceptable. Most of the participants being familiar with GeneXpert test ranked the Truenat results for TB at par with that of GeneXpert. It is important to point out that it is the substantiated accuracy of the Truenat machine in the detection of TB and also Rifampicin resistance coupled with its use in primary health centers and in hard-to-reach areas that made Truenat receive massive applause from healthcare workers and TB Program managers in Nigeria. This necessitated the attestation of the suitability of Truenat for our country's context by the participants bearing in mind the limited infrastructure in many of the health facilities in Nigeria. Thus, the participants in this study focused on areas where Truenat appear to have some comparative advantages over the GeneXpert. Although Nigeria has 507 installed GeneXpert machines, there is evidence that access to molecular WHO-recommended Diagnostic tests is still limited [11]. The healthcare workers being aware of this fact and the several limitations associated with the GeneXpert machine perceived Truenat as a welcome development and a support to GeneXpert in the fight against TB in Nigeria.

Although diagnostic accuracy studies involving Truenat were not conducted in-country before the roll-out, the recommendation by WHO based on results of a number of diagnostic accuracy tests, and the adoption of Truenat by the National Tuberculosis, Leprosy and Buruli ulcer Control Program (NTLCP) were adequately communicated to country stakeholders and end-users during the sensitization and trainings for roll-out of Truenat in Nigeria. All the participants remarked that good stakeholder engagement was fundamental in the successful implementation of Truenat in Nigeria. This engagement of stakeholders was not manifest at

the national level only but was a reality even at the health facility levels. Also, the process of engagement and sensitization of relevant stakeholders was observed to be a continuous process like when there was a perceived low utilization of Truenat. This would have resulted in the acceptance and perception of high reliability of the test results by the stakeholders. For example, New Tools implementing partners monitored utilization across sites and when utilization rate declined additional sensitization of providers/clinicians, laboratory staff and program managers were initiated to identify challenges and solicit opportunities for improvement. The consistent engagement proved to be essential to sustain Truenat throughput. Our study confirms that strong stakeholder engagement was indeed an important enabling factor in the implementation of Truenat for TB diagnosis, and will play a major role in scale up in Nigeria. This aligns with evidence that national health policies and plans that had good stakeholder engagement all through the policy cycle do have more effective implementation [12].

Another enabler of the Truenat implementation in Nigeria was training of health workers especially the laboratory staff that operated the Truenat machines. This was confirmed by the laboratory staff themselves. Training of health workers has remained a constant activity of the NTBLCP and these trainings have the support of Global Fund. The TB Control Program in Nigeria has its own unique recording and reporting tools that make training of healthcare workers necessary. A number of studies within Nigeria have demonstrated the need for training and retraining of healthcare workers involved in the delivery of TB services due to knowledge gaps [13–15]. Furthermore, a cluster randomized trial in southern Nigeria revealed an improvement in the knowledge of TB among frontline health workers after a training intervention. The study further suggested the integration of routine tuberculosis training to improve tuberculosis case finding in high burden countries [16].

The next identified enabler is the low operational cost associated with Truenat. Already, Truenat has been credited as being of relevance in the diagnosis of TB even in areas with limited infrastructure including the primary health centers [17]. Thus, what the implementing partners did during the implementation process included minor upgrade of already existing laboratories. To a large extent, Truenat machines unlike GeneXpert are less dependent on environmental conditions and suits resource limited health care settings [18]. However, it has to be pointed out that the 38 Truenat machines in use in Nigeria at present and the necessary reagents are all donations from USAID through the STOP TB Partnership. Perhaps, it is this great support from USAID that accounted for the low operational cost of the Truenat project in Nigeria.

It is not surprising that health worker related issues constituted the biggest barrier to the implementation of Truenat in Nigeria. These health worker issues included lack of laboratory staff to operate the Truenat machines, attrition of laboratory staff even after training on Truenat and lack of incentives. According to the WHO, even though the African region accounts for more than 24% of the global disease burden, the region holds 3% of the health workforce and have access to 1% of the world's financial resources even when grants and loans from foreign countries were included [19]. This constraint in human resource for health was further worsened by the COVID-19 pandemic [20]. In Nigeria as in all corners of the globe, there are fewer health workers in rural areas [19, 21]. It is these same rural communities of Nigeria where there is minimal health infrastructure and fewer health workers that were expected to host the Truenat machines. That could explain the position of the participants in this study.

The limited availability of personnel in hard-to-reach areas is exacerbated by the high attrition of health workers which further affected the implementation process. Over the course of implementation, several laboratory staff who received training accepted more lucrative positions which resulted in the need for continuous investments in recruitment of staff. It was also remarked that the inability to circumvent these departures through compensatory incentives

was a limiting factor. This was a significantly debilitating factor since the majority of peripheral sites had one laboratory staff that conducted all the laboratory tests. Due to ongoing staff attrition, there was the need to provide financial incentives so as to retain staff with the addition of Truenat testing which was not an affordable venture. This absence of incentive adversely affected the utilization rate of the Truenat machine at some sites as there was no motivation for the laboratory staff to take on the additional workload from Truenat. To circumvent the ongoing human resource challenges, USAID implementing partners hired, trained and provided stipends for ad-hoc Truenat laboratory staff across impacted sites.

A study in Lagos state has pointed out that the Lagos state TB program is plagued with lack of manpower [22], This could be the situation in other states of the federation as inadequate number of frontline TB healthcare workers in primary health centers especially in rural communities has been identified as one of the health system factors that adversely affect the TB control program in Nigeria. This is because poor remuneration and lack of incentives combine to hinder the retention of the services of community level officers [23]. Thus, there has been a call for the training of TB healthcare workers and payment of incentives as a way of attracting and retaining TB service providers [24].

It is in this light that most of the participants anticipated that human resource constraints will be the biggest challenge to the scale up of the use of Truenat in Nigeria. The implementing partners are fully aware of this limitation and have started the engagement of ad-hoc staff for the operation of Truenat machines. It is surprising that the laboratory staff did not focus on human resource constraints as a limitation to the scale up of use of Truenat instead their concern was the availability of consumables or reagents for the effective functioning of the Truenat machines. This was irrespective of the fact that some of the laboratory staff had not witnessed shortage of consumables in their various health facilities. This could be a demonstration of patient centered care on the part of the laboratory staff and the quest for continuity of service delivery on the part of the other healthcare workers and program managers in the National TB Program.

In addition to human resource challenges, lack of power supply was the next major challenge inhibiting Truenat implementation process. Despite possessing an in-built battery, the limited battery life required a more consistent power source in order to regularly recharge. Lack of consistent power is a challenge especially in sites with higher throughput/utilization. This is specifically highlighted because part of the success of the Truenat machine is its turnaround time (TAT) and downtime due to battery life can expand this. Some of the participants who were clinicians and laboratory staff called for the installation of solar panels to support the Truenat machines.

One of the participants pointed out the negative influence of poor security as a barrier to the scale up of the use of Truenat in Nigeria. This is not the first time that security concerns are having a negative impact on healthcare delivery and utilization of services. In a study in Enugu state, Nigeria, security concerns limited the use of primary health centers for maternal health services. This necessitated a call for the fencing of primary health centers in the state for effective service delivery [25]. This makes it imperative that issues related to security are tackled immediately and directly because of its spill-over effects on health.

To a large extent, healthcare workers in the Nigerian TB Control Program are innovative and this is commendable. When Truenat came into the country with an in-built battery, it had an edge over the GeneXpert machine. This was appreciated by the healthcare workers and program managers. Soon, the reality dawned on everyone that the in-built batteries have to be charged, they then reflected on the solar panel of TB LAMP. This informed the call by the participants to the manufacturers of Truenat machines to attach a solar panel to the Truenat machine as is the case with TB LAMP. This is expected to address the poor power supply as a

barrier to the implementation of Truenat. As at present, some of the Truenat machines in Nigeria already have solar panels attached for effective service delivery courtesy of the implementing partners.

All the suggestions to Molbio, the manufacturers of Truenat on improving the performance of Truenat signal a total acceptance of the Truenat technology by the healthcare workers and TB program managers. It could be said that the fundamentals of Truenat offered a solid foundation that required little amendments for perfection. First is the number of modules of which GeneXpert came to the rescue. In-fact one health worker conceptualized the 'seeing of Truenat as GeneXpert' once the number of modules associated with Truenat is increased to at least four. One request that remained critical to the future of Truenat in Nigeria from the views of participants is the need to validate the use of stool for the diagnosis of TB using Truenat. This was based on the concern of the participants for the diagnosis of childhood TB. In Nigeria, childhood TB notification is low [26, 27]. This necessitated the call to prioritize the diagnosis of childhood TB [28]. Moreover, health workers in Nigeria have identified the inability of children to produce sputum as one of the challenges to diagnosis of childhood TB [29], prompting the need to use stool for the diagnosis as is the case with GeneXpert at present. That makes the call by the participants in this study very crucial. Perhaps, as a sign of the acceptance of Truenat by the healthcare workers, all the laboratory staff who participated in the study expressed their willingness to use stool to test for TB using Truenat.

It could be said that the future of Truenat in Nigeria is very bright especially when the suggestions by the participants are taken into consideration. It is also important to point out that most of the suggestions by the participants in this study on how to improve Truenat have already received the approval of the manufacturers of the machine. For example, Truenat with four modules referred to as Quattro is currently available and also Truenat with an attached solar panel. Furthermore, there are plans by USAID/KNCV Tuberculosis Foundation Nigeria to validate the use of Truenat for TB testing using stool samples. Like one of the participants pointed out, the need for use of molecular test for the diagnosis of TB in Nigeria coupled with the suitability of Truenat for use in primary health centers and hard to reach areas make Truenat very relevant in the country. When one counts the gains of Truenat in community interventions like the community based active case finding especially in combination with a portable digital X-ray, it becomes obvious that Truenat has come to stay in Nigeria. This is because Truenat while used at point of care for TB diagnosis improves linkage to care [7]. Moreover, this approach has the capacity to reduce the risk of community transmission of TB [30]. Again, Truenat when used effectively in Nigeria could be the game changer in finding the missing TB cases in Nigeria.

This study took place in fourteen states of Nigeria that are part of USAID supported TB LON 1, 2&3 projects. Notwithstanding, these states have been identified as having varying levels of diagnostic gaps for TB evaluation and are spread across the six geo-political zones of the country. Consequently, the findings from this study could be a good representation of the perspective of healthcare workers and TB Program managers on the use of Truenat for TB diagnosis in Nigeria. To minimize bias, an external Consultant who is a university lecturer was engaged for the conduct of the study by KNCV Nigeria on behalf of USAID.

## Conclusion

Truenat has gained high acceptance among health workers and TB Program managers in Nigeria. The strong stakeholder engagement, perceived low operational cost and low infrastructural requirements have been a major boost for acceptance. Training should be sustained and measures put in place to ensure the retention of the trained Truenat laboratory staff

including the payment of incentives. Increasing the number of modules of Truenat machine, enabling the use of stool for TB diagnosis and having a solar panel attached to Truenat machine are essential.

## Supporting information

**S1 File. Transcripts.**
(DOCX)

## Acknowledgments

The authors acknowledge the support by the National Tuberculosis and Leprosy Control Program, State TB Program Managers across the 18 USAID TB-LON supported states in Nigeria, Tuberculosis Local Government Supervisors, Health care workers in the selected sites, and study participants for their contribution to the success of this study. We also appreciate the technical support and input from the Stop TB partnership introducing new tools project (iNTP) team and USAID Washington TB Division.

## Author Contributions

**Conceptualization:** Nkiru Nwokoye, Austin Ihesie, Jamiu Olabamiji, Kingsley Ochei, Rupert Eneogu, Debby Nongo, Aderonke Agbaje, Bethrand Odume.

**Data curation:** Nkiru Nwokoye, Austin Ihesie, Jamiu Olabamiji, Kingsley Ochei, Rupert Eneogu, Michael Umoren, Femi Odola, Debby Nongo, Omosalewa Oyelaran, Edmund Ndudi Ossai.

**Formal analysis:** Edmund Ndudi Ossai.

**Funding acquisition:** Austin Ihesie, Rupert Eneogu.

**Investigation:** Debby Nongo, Aderonke Agbaje, Bethrand Odume.

**Methodology:** Nkiru Nwokoye, Austin Ihesie, Jamiu Olabamiji, Kingsley Ochei, Rupert Eneogu, Wayne van Germert, Lucy Mupfumi.

**Project administration:** Austin Ihesie, Kingsley Ochei, Rupert Eneogu, Bethrand Odume.

**Resources:** Nkiru Nwokoye, Austin Ihesie, Jamiu Olabamiji, Kingsley Ochei, Rupert Eneogu, Michael Umoren, Femi Odola, Debby Nongo, Aderonke Agbaje, Bethrand Odume, Omosalewa Oyelaran, Elom Emeka, Chukwuma Anyaike.

**Supervision:** Nkiru Nwokoye, Austin Ihesie, Jamiu Olabamiji, Kingsley Ochei, Rupert Eneogu, Michael Umoren, Femi Odola, Debby Nongo, Aderonke Agbaje, Bethrand Odume, Omosalewa Oyelaran.

**Writing – original draft:** Nkiru Nwokoye, Austin Ihesie, Jamiu Olabamiji, Kingsley Ochei, Rupert Eneogu, Edmund Ndudi Ossai.

**Writing – review & editing:** Nkiru Nwokoye, Austin Ihesie, Jamiu Olabamiji, Kingsley Ochei, Rupert Eneogu, Michael Umoren, Femi Odola, Debby Nongo, Aderonke Agbaje, Bethrand Odume, Omosalewa Oyelaran, Wayne van Germert, Lucy Mupfumi, Elom Emeka, Chukwuma Anyaike, Sarah Cook Scalise, Edmund Ndudi Ossai.

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
