## [Decision Letter · Decision Letter 0]

11 Jul 2024

PONE-D-24-17897Exploring the perspectives of healthcare workers and Program managers on the use of Truenat as a new tool for TB and DR-TB diagnosis in Nigeria: A qualitative studyPLOS ONE

Dear Dr. Ossai,

Thank you for submitting your manuscript to PLOS ONE. After careful consideration, we feel that it has merit but does not fully meet PLOS ONE’s publication criteria as it currently stands. Therefore, we invite you to submit a revised version of the manuscript that addresses the points raised during the review process.

**ACADEMIC EDITOR: **

METHODS

What about study setting/ study area? Please ensure that it is inputted.

Location of the 38 Truenat sites; States and LGA,s.  A map will be suitable for that.

Kindly state the duration of the study period.

Inclusion criteria was stated earlier on (line 137-141) and exclusion criteria was “No interview was repeated” or inability to participate in the interview twice. Please, state these points in your manuscript as inclusion and exclusion criteria.

Was there a point you reach “data saturation point” where no new information was obtained from the KII or the focus groups discussion? Please, state it in your methodology if data saturation point was reached or in the DISCUSSION section if it was not reached as part of the study limitation.

Kindly state the type of coding that was done: deductive or inductive method of coding in your methodology?

Methods that the authors use to minimize bias should be stated.

RESULTS

Interviewer characteristics (187-190) should be part of the methods and not results.

When quoting the statements of the participants use L1, L2------ for lab participants, K1, K2…… for KIIs. You need to make participants anonymous as much as possible.

Lines 258 -264 should be part of Discussion and not RESULTS….“It is also important to point out that the sensitization meetings by implementing partners were a continuous process especially when there was the need to increase the utilization rate of Truenat. For example, New Tools implementing partners monitored utilization across sites and when utilization rate declined additional sensitization of providers/clinicians, laboratory staff and program manager were initiated to identify challenges and solicit opportunities for improvement. The consistent engagement proved to be essential to sustain Truenat throughput.”

Lines 300-321… points from the findings should be clearly laid as results while the expansion on circumstances should be part of discussion.

Please, ensure that you create a table in the result section with columns having the headings; Themes, Subthemes, Codes generated and Repetition of codes by participants. This will make your work smart and easier to understand.

Ensure that the major topics are designated as Themes in your manuscript and the subtopics under each theme as Subtheme. For easier comprehension of your results.

DISCUSSION

Remove the heading “limitation” and just make the last paragraph under the DISCUSSION as limitations of the study.

We look forward to receiving your revised manuscript.

Kind regards,

Ayi Vandi Kwaghe, D.V.M., M.V.Sc., P.G.D.E. Ph.D., MPH, FETP

Academic Editor

PLOS ONE

 [The study was funded by the United States Agency for International Development (USAID) through the Stop TB partnership introducing new tools project (iNTP). ].  

Additional Editor Comments (if provided):

Reviewers' comments:

Reviewer's Responses to Questions

**Comments to the Author**

1. Is the manuscript technically sound, and do the data support the conclusions?

Reviewer #1: Partly

Reviewer #2: Yes

2. Has the statistical analysis been performed appropriately and rigorously? 

Reviewer #1: No

Reviewer #2: N/A

3. Have the authors made all data underlying the findings in their manuscript fully available?

Reviewer #1: No

Reviewer #2: Yes

4. Is the manuscript presented in an intelligible fashion and written in standard English?

Reviewer #1: No

Reviewer #2: Yes

5. Review Comments to the Author

Reviewer #1: The authors have conducted a qualitative research in a unique area that will certainly add and contribute significantly to the body of knowledge. The conduct (methods) and presentation (English and grammar), however have not been rigorously attended to. It is advised for authors to take the issues raised in this review to facilitate a timely decision.

Reviewer #2: It was qualitative study on healthcare workers and program managers on use of Truenat for TB

diagnosis . Researchers worked on 6 quality issues, a good work definitely.

Hence i have some observation which are as follows

1. Though mentioned in methodology , use of statistical tools seems less.

2. Line 193: Most of the participants, 54.5%, were males, better to write more than half

3. Line 194 More than two thirds, please mention percentage in bracket

4. In methodology please make it clear about whether same or different information was tried to explore from FGD and KII

5. in discussion , it will be better to mention similarities or difference in same pattern of studies

6. PLOS authors have the option to publish the peer review history of their article (what does this mean?). If published, this will include your full peer review and any attached files.

Reviewer #1: **Yes: **Taiwo A Obembe

Reviewer #2: No

---

## [Author Response · Author response to Decision Letter 0]

7 Aug 2024

Review comments and response from Authors 2

Comment

We note that this data set consists of interview transcripts. Can you please confirm that all participants gave consent for interview transcript to be published?

Author response

I confirm that all the participants consented for the interview transcripts to be published

Comment

If they DID provide consent for these transcripts to be published, please also confirm that the transcripts do not contain any potentially identifying information (or let us know if the participants consented to having their personal details published and made publicly available). We consider the following details to be identifying information:

Author response

Thanks for the observation. The participants DID provide consent for these transcripts to be published. The interview transcripts have been reviewed again and all potentially identifying information have been removed.

Comment

 We note that your Data Availability Statement is currently as follows: "All relevant data are within the manuscript and its Supporting Information files"

Author response

All relevant data related to the manuscript is uploaded as a Supporting Information file named ‘Transcripts.’

---

## [Decision Letter · Decision Letter 1]

24 Sep 2024

PONE-D-24-17897R1Exploring the perspectives of healthcare workers and Program managers on the use of Truenat as a new tool for TB and DR-TB diagnosis in Nigeria: A qualitative studyPLOS ONE

Dear Dr. Ossai,

Thank you for submitting your manuscript to PLOS ONE. After careful consideration, we feel that it has merit but does not fully meet PLOS ONE’s publication criteria as it currently stands. Therefore, we invite you to submit a revised version of the manuscript that addresses the points raised during the review process.

**ACADEMIC EDITOR: **Reviewers comments should be effected and where the authors have contrary views should be explained.=============================

We look forward to receiving your revised manuscript.

Kind regards,

Ayi Vandi Kwaghe, D.V.M., M.V.Sc., P.G.D.E. Ph.D., MPH, FETP

Academic Editor

PLOS ONE

Journal Requirements:

Reviewers' comments:

Reviewer's Responses to Questions

**Comments to the Author**

1. If the authors have adequately addressed your comments raised in a previous round of review and you feel that this manuscript is now acceptable for publication, you may indicate that here to bypass the “Comments to the Author” section, enter your conflict of interest statement in the “Confidential to Editor” section, and submit your "Accept" recommendation.

Reviewer #1: (No Response)

Reviewer #2: All comments have been addressed

Reviewer #3: (No Response)

2. Is the manuscript technically sound, and do the data support the conclusions?

Reviewer #1: Yes

Reviewer #2: Yes

Reviewer #3: Yes

3. Has the statistical analysis been performed appropriately and rigorously? 

Reviewer #1: Yes

Reviewer #2: Yes

Reviewer #3: N/A

4. Have the authors made all data underlying the findings in their manuscript fully available?

Reviewer #1: Yes

Reviewer #2: Yes

Reviewer #3: Yes

5. Is the manuscript presented in an intelligible fashion and written in standard English?

Reviewer #1: No

Reviewer #2: Yes

Reviewer #3: Yes

6. Review Comments to the Author

Reviewer #1: Some of the concerns raised have not been fully addressed. The yellow highlights on the re-submitted corrected version is good. Authors should kindly endeavor to refer to all queries that are un-addressed and provide a correction, a refute or an explanation for the points that remain unaddressed.

Reviewer #2: (No Response)

Reviewer #3: The author said the FGD and KII recordings were double transcribed immediately. It would have been good to know who did the transcription. Was it done by one person or two persons and then compared? Again, it would be important to know if the participants consented to the discussions and KII being recorded?

The Conceptual framework for this study is depicted as a roof (Acceptance and use of the Truenat machine) resting on six pillars (Learnability, Willingness, Suitability, Satisfaction, Efficiency, and Effectiveness). Unfortunately, the themes pursued in the FGDs and KIIs did not specifically pursue these pillars though one could infer from the issues raised that they tangentially made reference to them. It would have made the framework very relevant for those pillars to be the issues explored in the FGDs and KIIs so we see how this framework directly explained the acceptance and utilization of the Truenat machine in Nigeria. The way the framework was applied almost made it irrelevant to the study.

There were a few editorial errors on the manuscript that needs to be corrected. For example, in line 162, there should be a fullstop after themes, an the next sentence should start with a capital letter; In line 180, insert a comma (punctuation) after acceptance, then delete and and line 303, insert 'of' after recruitment.

7. PLOS authors have the option to publish the peer review history of their article (what does this mean?). If published, this will include your full peer review and any attached files.

Reviewer #1: **Yes: **Taiwo A Obembe

Reviewer #2: No

Reviewer #3: **Yes: **Prof Lawrence Ulu Ogbonnaya

---

## [Author Response · Author response to Decision Letter 1]

21 Oct 2024

EDITOR COMMENTS AND RESPONSE FROM AUTHORS

Editor comment

1. We note that your Data Availability Statement is currently as follows: "All relevant data are within the manuscript and its Supporting Information files"

Author response

I confirm that all relevant data are within the manuscript and its Supporting Information file titled Transcripts_REVIEWED.

Editor comment

2. We note that Figure 1 in your submission contain [map/satellite] images which may be copyrighted. All PLOS content is published under the Creative Commons Attribution License (CC BY 4.0), which means that the manuscript, images, and Supporting Information files will be freely available online, and any third party is permitted to access, download, copy, distribute, and use these materials in any way, even commercially, with proper attribution. For these reasons, we cannot publish previously copyrighted maps or satellite images created using proprietary data, such as Google software (Google Maps, Street View, and Earth). For more information, see our copyright guidelines: http://journals.plos.org/plosone/s/licenses-and-copyright.

Author response

Thanks for the observation. Figure 1 has been deleted.

---

## [Decision Letter · Decision Letter 2]

8 Dec 2024

Exploring the perspectives of healthcare workers and Program managers on the use of Truenat as a new tool for TB and DR-TB diagnosis in Nigeria: A qualitative study

PONE-D-24-17897R2

Dear Dr. Ossai,

We’re pleased to inform you that your manuscript has been judged scientifically suitable for publication and will be formally accepted for publication once it meets all outstanding technical requirements.

Kind regards,

Ayi Vandi Kwaghe, D.V.M., M.V.Sc., P.G.D.E. Ph.D., MPH, FETP

Academic Editor

PLOS ONE

Additional Editor Comments (optional):

Reviewers' comments:

Reviewer's Responses to Questions

**Comments to the Author**

1. If the authors have adequately addressed your comments raised in a previous round of review and you feel that this manuscript is now acceptable for publication, you may indicate that here to bypass the “Comments to the Author” section, enter your conflict of interest statement in the “Confidential to Editor” section, and submit your "Accept" recommendation.

Reviewer #1: All comments have been addressed

Reviewer #2: All comments have been addressed

2. Is the manuscript technically sound, and do the data support the conclusions?

Reviewer #1: Yes

Reviewer #2: Yes

3. Has the statistical analysis been performed appropriately and rigorously? 

Reviewer #1: Yes

Reviewer #2: Yes

4. Have the authors made all data underlying the findings in their manuscript fully available?

Reviewer #1: Yes

Reviewer #2: Yes

5. Is the manuscript presented in an intelligible fashion and written in standard English?

Reviewer #1: Yes

Reviewer #2: Yes

6. Review Comments to the Author

Reviewer #1: The comments have been satisfactorily addressed. The authors have done a thorough review and corrected all that needs to be corrected. The manuscript is recommended for publication.

Reviewer #2: (No Response)

7. PLOS authors have the option to publish the peer review history of their article (what does this mean?). If published, this will include your full peer review and any attached files.

Reviewer #1: **Yes: **Taiwo Obembe

Reviewer #2: No

---

## [Editor Report · Acceptance letter]

16 Dec 2024

PONE-D-24-17897R2 

PLOS ONE

Dear Dr. Ossai, 

I'm pleased to inform you that your manuscript has been deemed suitable for publication in PLOS ONE. Congratulations! Your manuscript is now being handed over to our production team.

Kind regards, 

on behalf of

Dr. Ayi Vandi Kwaghe 

Academic Editor

PLOS ONE